# Saliva urea nitrogen for detection of kidney disease in adults: A meta-analysis of diagnostic test accuracy

Reyna Kumaran[1], Mona Mohamed Ibrahim Abdalla[1]*, Brinnell Annette Caszo[1], Sushela Devi Somanath[2]

1 Human Biology Department, School of Medicine, IMU University (Formerly known as International Medical University), Kuala Lumpur, Malaysia, 2 Pathology Department, School of Medicine, IMU University (Formerly known as International Medical University), Kuala Lumpur, Malaysia

* monamohamed@imu.edu.my

## Abstract

### Background

Kidney disease affects millions globally, especially in low and middle-income countries where access to diagnostic testing is limited. Saliva urea nitrogen (SUN) has been proposed as a simple, non-invasive alternative to traditional serum-based diagnostics.

### Objective

This study aimed to evaluate the diagnostic accuracy of SUN for detecting kidney disease in adults through a systematic review and meta-analysis.

### Methods

This review adhered to the PRISMA-DTA guidelines. A comprehensive search of five databases was conducted without language or date restrictions. Study quality was assessed using the QUADAS-2 tool. STATA version 17 was used for analysis. A random-effects model was used to estimate pooled sensitivity, specificity, and diagnostic odds ratios (DOR). Subgroup analysis was conducted based on the reference test used (serum creatinine or blood urea nitrogen). Heterogeneity was assessed using the $I^2$ statistic, and meta-regression explored sources of heterogeneity.

### Results

Seven studies (n = 1,933) met the inclusion criteria. In the serum creatinine (sCr) subgroup (2 studies), SUN showed pooled sensitivity of 0.44 (95% CI: 0.38–0.49), specificity 0.96 (95% CI: 0.95–0.98), DOR 18.89 (95% CI: 15.19–23.57), and AUC ~0.90. In the blood urea nitrogen (BUN) subgroup (5 studies), sensitivity was 0.83 (95%

**Data availability statement:** All relevant data are within the paper and its Supporting Information files.

**Funding:** International Medical University (IMU) in Malaysia [ID: CSc-Sem6(08)2022]. The funders had no role in study design, data collection and analysis, decision to publish, or preparation of the manuscript.

**Competing interests:** The authors have declared that no competing interests exist.

CI: 0.69–0.91), specificity 0.88 (95% CI: 0.78–0.94), DOR 37 (95% CI: 15–91), and AUC 0.93. Heterogeneity was moderate in the BUN subgroup (bivariate $I^2 = 51\%$), with 42% of variability attributed to threshold effects. Meta-regression identified study country ($p = 0.01$), and reference test used ($p = 0.02$) as contributors to heterogeneity in sensitivity, while comorbidity ($p = 0.001$) significantly affected specificity.

## Conclusion

SUN shows high diagnostic specificity and a good overall accuracy, particularly when compared to BUN, and may serve as a practical non-invasive screening tool in low- resource settings. While heterogeneity was present, SUN remains a promising diagnostic alternative and warrants further validation in diverse clinical populations.

## Introduction

The global incidence of both acute kidney disease (AKD) and chronic kidney disease (CKD) has been steadily rising in recent years, contributing significantly to morbidity, mortality, and escalating healthcare costs [1]. CKD, defined by a sustained impairment in kidney function lasting more than three months, is stratified into five stages, reflecting the severity of kidney damage and functional decline [2]. In contrast to the gradual progression of CKD, AKD is characterized by a rapid deterioration in kidney function, necessitating immediate medical intervention [3]. It is estimated that 5% to 7% of the global population experiences mild to moderate CKD, predominantly caused by non-communicable diseases such as type 2 diabetes and hypertension. These conditions disproportionately affect minority and socioeconomically disadvantaged groups, highlighting a critical gap in access to necessary healthcare resources [4].

Early detection and accurate diagnosis of kidney dysfunction are critical for initiating timely interventions that can slow disease progression, improve patient outcomes, and reduce healthcare expenditures [5]. Conventional diagnostic markers for kidney function include serum creatinine (sCr) and blood urea nitrogen (BUN), which are used to estimate glomerular filtration rate (GFR), the gold standard for assessing renal function [6]. However, these tests require venipuncture, an invasive procedure that may cause discomfort, and carries a risk of infection. Furthermore, access to such diagnostic facilities is often limited in low-resource settings, highlighting the need for non-invasive, accessible, and cost-effectives.

Recent years have seen the emergence of several novel biomarkers proposed for the diagnosis and monitoring of kidney disease. For instance, hsa_Circ_0072463 has shown a strong correlation with serum creatinine levels in both septic and non-septic acute kidney injury (AKI) patients [7]. Similarly, serum indole-3-aldehyde, a gut microbiota-derived metabolite, has demonstrated renoprotective properties and is negatively correlated with creatinine levels in CKD patients [8]. Additional studies have associated progressive tubulointerstitial fibrosis to metabolites like 1-methoxypyrene and increased expression of aryl hydrocarbon receptor pathway genes in models of ureteric obstruction [9]. Moreover, urinary biomarkers like vanin-1

and neutrophil gelatinase-associated lipocalin have shown promise as early biomarkers of renal tubular damage in hypertensive rats [10]. While these biomarkers are promising, many lack standardization and robust clinical validation, reinforcing the importance of exploring simpler and scalable alternatives such as SUN [11].

Saliva has gained attention as a valuable medium for biomarker analysis due to its non-invasive collection, ease of handling, and potential for monitoring various diseases, including kidney disorders [12]. Salivary diagnostics offer a patient-friendly alternative to blood- based testing, enabling frequent monitoring without the need for trained personnel or specialized equipment. Because the composition of saliva reflects many physiological processes, it serves as a rich source of biological markers for disease detection and monitoring [13].

Urea nitrogen, a metabolic byproduct of protein catabolism, is primarily excreted by the kidneys [14]. Elevated levels of BUN are indicative of impaired kidney function. Saliva urea nitrogen, which mirrors serum urea levels due to the free diffusion of urea across the salivary gland epithelium, has emerged as a potential surrogate marker for kidney function. Thus, SUN levels may provide a non-invasive method for assessing kidney health by reflecting the uremic state of the individual [15]. In healthy individuals, SUN levels typically range from 5 to 14 mg/dL, though these values may vary slightly depending on population and methodology. In contrast, SUN levels between 35–54 mg/dL are frequently observed in individuals with kidney dysfunction [16–17].

This meta-analysis aims to systematically review and synthesize existing research on the diagnostic accuracy of SUN for detecting kidney disease in adults.

## Methods

This study adhered to the guidelines outlined in the Preferred Reporting Items for Systematic Reviews and Meta-Analyses for Diagnostic Test Accuracy (PRISMA-DTA)." The completed PRISMA-DTA checklist is available in S1 Table. The protocol was approved by the Joint Committee on Research and Ethics at the International Medical University (IMU) in Malaysia [ID: CSc-Sem6(08)2022] and is available upon request from the corresponding author. The proposal for this study was not registered.

### Search strategy

A comprehensive search conducted using electronic databases including Scopus, PubMed, ScienceDirect, Web of Science, and PubMed Central. The search utilized relevant keywords such as "renal disease," "renal failure," "kidney disease," "saliva urea nitrogen," and "saliva urea," combined using Boolean operators "AND" and "OR" to refine results. Detailed search strategies for each database are provided in S2 Table. No restrictions were applied regarding language, year of publication, or country of origin. Studies published up to January 2025 were considered. Additionally, the reference lists of relevant studies and systematic reviews were manually searched to ensure comprehensive coverage. All search results from the databases were imported into EndNote 2020 software for screening.

### Eligibility criteria and study selection

Studies were selected based on predefined inclusion and exclusion criteria, following the "PIRD" framework (Population, Index test, Reference test, Diagnosis of interest), as recommended for diagnostic test accuracy (DTA) systematic reviews [18]. Eligible studies included adults aged 18 years or older, regardless of gender, ethnicity, or geographical location. The index test had to be the SUN test, regardless of the manufacturer. The reference test included measurement of GFR, the gold standard for diagnosing kidney disease, or alternative tests such as sCr or BUN. Studies assessing either AKD or CKD, as defined by the primary research, were eligible.

To be included, studies were required to report diagnostic accuracy of the SUN test, providing sufficient information to construct a 2x2 contingency table (True Positive [TP], False Negative [FN], True Negative [TN[, and False Positive [FP]) [19–20]. Exclusion criteria included studies focused on pediatric population, as well as reviews, commentaries, protocols, conference abstracts, case reports, and letters to editors.

## Data extraction

Two investigators independently screened the titles and abstracts of the retrieved citations, selected full-text articles that met the inclusion criteria. Data extraction was performed using a pre-designed data extraction. The following key information was extracted from each study: first author, publication year, country of origin, participant characteristics (age, gender), study design, sample size, setting, study period, diagnostic tests used (index and reference), blinding procedures for both tests, and diagnostic performance metrics (TP, FP, FN, TN). Disagreements were resolved through discussion with a third investigator to reach a consensus.

## Risk of bias assessment

The methodological quality of the included studies was independently assessed by two researchers using the revised version of the Quality Assessment of Diagnostic Accuracy Studies (QUADAS-2) tool, which is specifically designed to evaluate DTA studies [21]. This tool assesses the risk of bias and applicability concerns across four domains: patient selection, index test, reference standard, and flow and timing. Each domain was evaluated using signaling questions, with responses classified as "yes," "no," or "unclear." Based on these assessments, each domain was rated as having "low," "unclear," or a "high" risk of bias. Any discrepancies were resolved by consensus with a third investigator.

## Statistical analysis and data synthesis

The primary test performance indicators were sensitivity and specificity, each with corresponding 95% confidence intervals (CIs). Sensitivity was defined as the proportion of individuals with the target condition who tested positive, while specificity referred to the proportion of individuals without the condition who tested negative [20,22,23]. These values were expressed as proportions (0–1) or percentages. The diagnostic odds ratio (DOR), which quantifies the odds of the target condition being present in those who test positive versus those who test negative, was also calculated [20].

Pooled analyses were performed when at least two eligible studies were available. Pooled sensitivity and specificity with 95% CIs, were estimated using a random-effects model to account for between study-heterogeneity and were visually presented in a forest plot. Heterogeneity was quantified using the $I^2$ statistic, with values > 50% indicating substantial heterogeneity[20].

The receiver operating characteristic (ROC) curve was used to visually represent test performance, plotting sensitivity (true-positive rate) against 1-specificity (false-positive rate) [20]. Summary receiver operating characteristic (SROC) curves were generated to illustrate overall test accuracy across various thresholds, following the methodology described by Macaskill et al. (2010). The area under the SROC curve (AUC), was used as a summary measure of diagnostic accuracy: a value of 1.0 indicates perfect accuracy, 0.5 suggests a non-informative test, and values between 0.5 and 0.7, 0.7 and 0.9, and > 0.9 were interpreted as indicating low, moderate, and high accuracy, respectively [20].

Meta-regression analyses were conducted to explore potential sources of heterogeneity by examining covariates such as sample size, risk of bias, study design, reference test used, and blinding status for both index and reference tests. A p-value < 0.1 was considered indicative of significant heterogeneity [22]. Publication bias was not assessed, as it is not recommended for DTA studies [24].

All analyses were performed using STAT version 17 (StataCrop, College Station, TX). For subgroups with four or more studies, the MIDAS package was used to perform bivariate diagnostic meta-analysis. For subgroups with only two studies, the metadta package was employed to generate pooled estimates under restricted conditions. In such subgroups, heterogeneity metrics could not always be reliably calculated.

To address methodological heterogeneity, the diagnostic performance of SUN was evaluated separately based on the reference test used, either sCr or BUN. This stratified approach allowed for a clearer understanding of SUN's accuracy in different diagnostic contexts. Pooled estimates of sensitivity, specificity, and AUC were calculated independently for each subgroup.

## Results

### Study selection

The study selection process is illustrated in Fig 1, which presents the PRISMA flow diagram. Initial database searches yielded 2,871 citations (PubMed: 493; PubMed Central: 717; Scopus: 877; Science Direct: 715; Web of Science: 69). After removing 348 automatically recognized duplicates and excluding 2,459 citations based on relevance, 64 records were selected for further screening. Following title and abstract screening, 13 full-text articles were evaluated for eligibility. Seven studies met the inclusion criteria and were included in the final review and analysis [16,17,25–29]. A detailed list of all 2,523 screened studies, including titles and reasons for exclusion, is provided in S1 File.

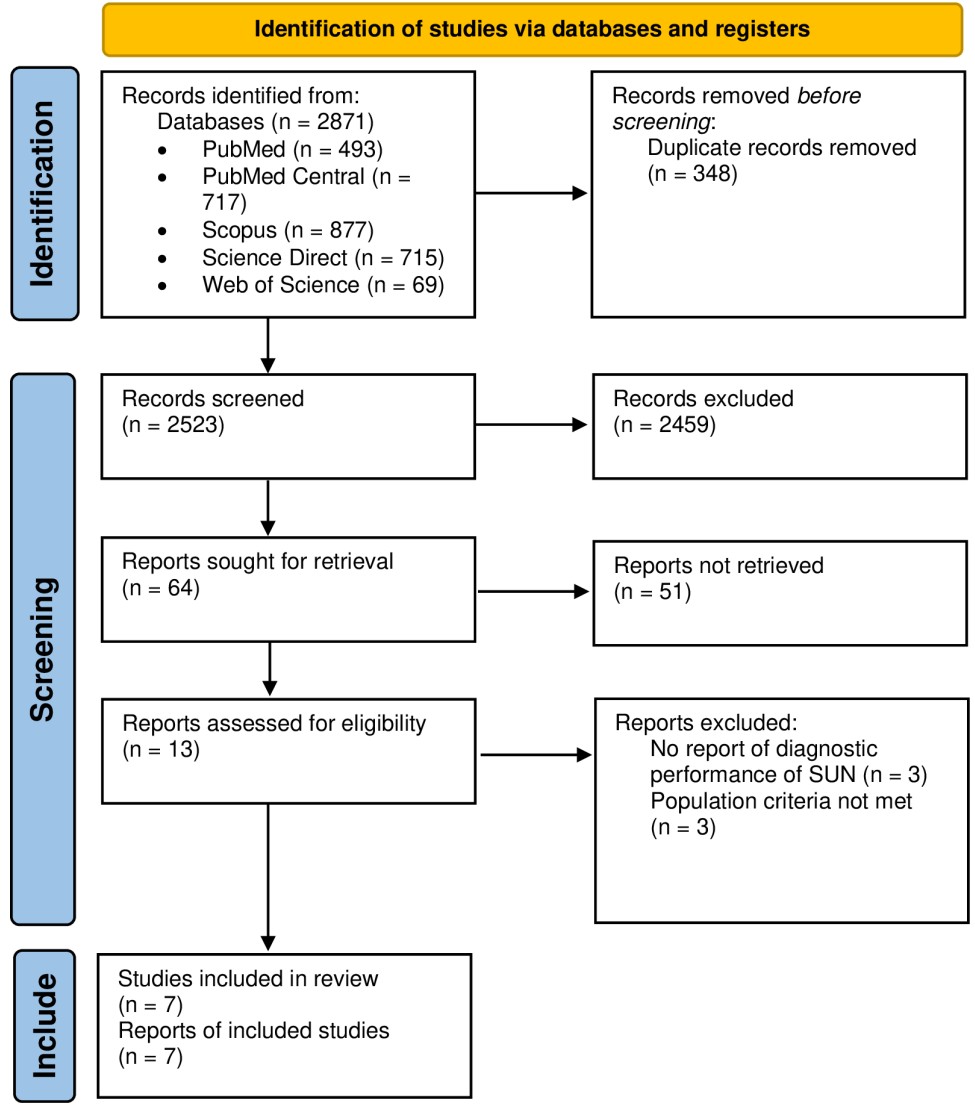

**Fig 1. PRISMA flow diagram.**

## Characteristics of included studies

The seven studies included in this meta-analysis were published between 2011 and 2020. Main characteristics are summarized in Table 1. Two studies (28.6%) were cross-sectional, and five (71.4%) were prospective. Five studies (71.4%) were conducted in a single country, three in Malawi, one in Brazil, and one in Angola. The remaining two (28.6%) were multi-country studies: one in Austria and the United States of America (USA), and another in Brazil and the USA. Notably, no studies originated from Asia, highlighting a geographic imbalance. Detailed extracted data from the included studies are presented in S2 File.

## Saliva sampling and SUN diagnostic thresholds

Table 2 summarizes the saliva sampling protocols and the diagnostic thresholds used in included studies. All seven studies used 50 µl of unstimulated saliva collected and applied to a colorimetric SUN dipstick (Integrated Biomedical Technology). Most studies (57%) used a diagnostic threshold of > 14 mg/dL. Reference tests varied across the studies: sCr was used in two studies (Evans et al., 2018; Evans et al., 2020), and BUN in the remaining five (Silva, 2014; Silva, 2018; Evans, 2017; Raimann, 2011; Raimann, 2016).

## Methodological quality of included studies

The QUADAS-2 assessment is summarized in Fig 2. Two studies (Calice-Silva 2018; Evans et al. 2020) had a low risk of bias across most domains. In contrast, three studies (Raimann et al. 2011; Raimann et al. 2016; Silva 2014) showed 'unclear risk of bias due to uncertainties in patient selection and recruitment. One study (Raimann et al. 2016) was rated high risk of bias in the 'flow and timing' domain, indicating in sample collection or other procedural timing.

In terms of applicability, four studies (Calice-Silva 2014; Evans et al. 2017; Evans et al. 2018; Evans et al. 2020) had low concerns across all domains. Silva (2014) had unclear concern regarding the reference standard, and Raimann

**Table 1. Characteristics of included studies.**

| Author | Year | Country | Disease condition | Study design | Sample size | Mean age | % male | Index test | Ref test | TP | FP | FN | TN |
|---|---|---|---|---|---|---|---|---|---|---|---|---|---|
| **Raimann** | 2011 | Austria USA | CKD HD | CS | 68 (34 CKD, 34 HD) | 61 | 54 | SUN | BUN | 80 | 3 | 13 | 24 |
| **Silva** | 2014 | Brazil | AKI | CS | 44 (44 AKI) | 59.5 | 42 | SUN | BUN | 10 | 7 | 1 | 26 |
| **Raimann** | 2016 | Brazil USA | AKI | Pros | 37 (6 control, 31 AKI) | 60 | 59.5 | SUN | BUN | 26 | 1 | 1 | 9 |
| **Evans** | 2017 | Malawi | AKI AKD CKD | Pros | 742 (596 control, 114 AKI, 26 AKD, 6 CKD) | 41 | 56 | SUN | BUN | 105 | 77 | 41 | 519 |
| **Silva** | 2018 | Angola | AKI AKD | Pros | 86 (59 control, 15 AKI, 12 AKD) | 21.5 | 71 | SUN | BUN | 18 | 1 | 9 | 58 |
| **Evans** | 2018 | Malawi | AKI AKD | Pros | 301 (262 control, 23 AKI, 16 AKD) | 25.9 | 0 | SUN | sCr | 5 | 7 | 34 | 255 |
| **Evans** | 2020 | Malawi | AKI AKD CKD | Pros | 655 (173 control, 130 AKI, 343 AKD, 10 CKD) | 38 | 44 | SUN | sCr | 143 | 15 | 339 | 158 |

SUN: Saliva Urea Nitrogen; BUN: Blood Urea Nitrogen; sCr: Serum Creatinine; CKD: Chronic Kidney Disease; AKI: Acute Kidney Injury; AKD: Acute Kidney Disease; HD: Haemodialysis; TP: True positive; FP: false positive; FN: false negative; TN: true negative; CS: cross-sectional study; Pros: Prospective study; Ref test: reference test; USA: United States of America

**Table 2. Saliva sampling methods and diagnostic thresholds used in included studies.**

| Author, Year | Saliva sampling method | Time for blood sampling in relation to saliva sampling | Type of saliva sampling | Type of blood sampling | Threshold for diagnosis for index test | Threshold for diagnosis for reference test |
|---|---|---|---|---|---|---|
| **Raimann 2011** [25] | 1 to 2 ml of unstimulated saliva was collected from all subjects in a plastic cup and allowed to separate in a liquid and foamy phase over a period of 1–3 minutes. 50 µl of liquid saliva was transferred to the test pad of the SUN dipstick. | Saliva was collected within 10 minutes of blood collection | Colorimetric SUN dipstick (Integrated Biomedical Technology) | BUN | SUN level of 15–24 mg/dL | BUN > 25 mg/dL |
| **Silva, 2014** [28] | 1 to 2 ml of unstimulated saliva was collected from all subjects in a plastic cup and allowed to separate in a liquid and foamy phase over a period of 1–3 minutes. 50 µl of liquid saliva was transferred to the test pad of the SUN dipstick. | Saliva was collected within 4 hours of blood collection | Colorimetric SUN dipstick (Integrated Biomedical Technology, Elkhart, IN, USA) | BUN | SUN level 35–54 mg/dL | BUN > 47 mg/dL |
| **Raimann 2016** [26] | Unstimulated Saliva was collected in a plastic cup. 50 µl of saliva was used to moisten the SUN dipstick. | Saliva was collected within 4 hours of blood collection | Colorimetric SUN dipstick (Integrated Biomedical Technology, Elkhart, Ind., USA) | BUN | NA | NA |
| **Evans 2017** [27] | Unstimulated Saliva was collected in a plastic cup. 50 µl of saliva was used to moisten the SUN dipstick. | Saliva and blood samples were collected simultaneously | Colorimetric SUN dipstick (Integrated Biomedical Technology) | sCr by the Jaffe method and BUN either by Flexor Junior Clinical Chemistry Analyzer [Vital Scientific, Dieren, The Netherlands] or by Mindray Chemistry Analyzer BS-120 [Shenzhen Mindray Bio-Medical Electronics Company, Shenzhen, China]) | A SUN > 14 mg/dL | sCr > 90 mmol/L (women), > 104 mmol/L (men) |
| **Silva 2018** [16] | Unstimulated Saliva was collected in a plastic cup. 50 µl of saliva was used to moisten the SUN dipstick. | Saliva and blood samples were collected simultaneously | Colorimetric SUN dipstick | BUN and sCr tests, 3–5 mL of blood was collected from a peripheral vein & processed in automated devices (Vital Scientific Flexor E180 and Flexor E450) | NA | Mean admission values: sCr = 5.38 mg/dL, BUN = 99.4 mg/dL; thresholds not explicitly defined. |
| **Evans 2018** [29] | Unstimulated Saliva was collected in a plastic cup. 50 µl of saliva was used to moisten the SUN dipstick. | Saliva and blood samples were collected simultaneously | Colorimetric (Integrated Biomedical Technology, Elkhart, IN) | sCr was measured by the Jaffe method, using either a Flexor Junior Clinical Chemistry Analyzer (Vital Scientific, Dieren, Netherlands) or a Mindray Chemistry Analyzer BS-120 (Shenzhen Mindray Bio-Medical Electronics Company, Shenzhen, China). | A SUN result of >14 mg/dL | CKD: eGFR < 60 mL/min/1.73 m² or known history; AKD: albuminuria or eGFR < 75; AKI: sCr change ≥ 0.3 mg/dL in 48h or 1.5 × baseline |
| **Evans 2020** [17] | Unstimulated Saliva was collected in a plastic cup. 50 µl of saliva was used to moisten the SUN dipstick. | Saliva and blood samples were collected simultaneously | Colorimetric SUN dipstick (Integrated Biomedical Technology, IN, USA) | sCr was measured at the POC using the StatSensor Xpress Creatinine device (Nova Biomedical Cooperation, Waltham, Massachusetts, USA) | A SUN result of >14 mg/dL | |

SUN: Saliva Urea Nitrogen; BUN: Blood Urea Nitrogen; sCr: serum creatinine; NA: not available/not reported; AKD: Acute Kidney Disease; AKI; Acute Kidney Injury; CKD; Chronic Kidney Disease; eGFR: estimated Glomerular Filtration Rate.

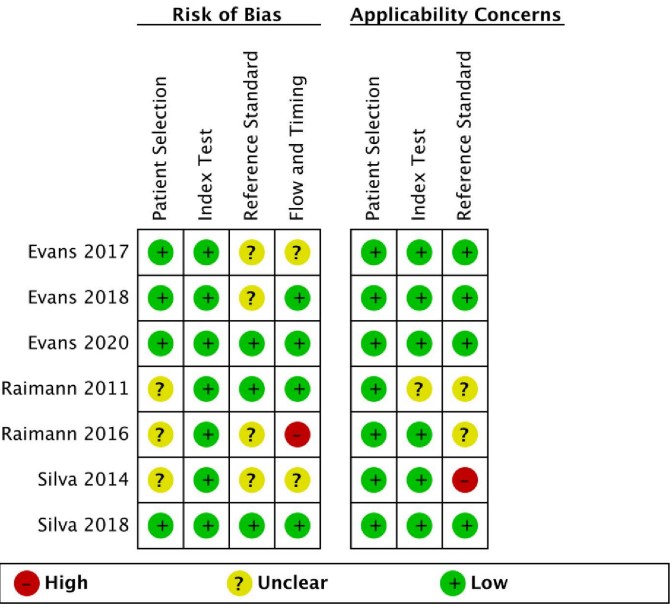

**Fig 2. Methodological quality of included studies.**

(2016) showed high concern, indicating potential issues with the relevance or application of the reference standard used. A detailed breakdown of the risk of bias and applicability assessments for each study is available in S3 Table.

### Test performance

The diagnostic test performance of individual studies using SUN is summarized in S4 Table.

### Saliva urea nitrogen performance using serum creatinine as the reference test

Two studies (Evans 2018; Evans 2020) used sCr as the reference test. As shown in Fig 3, the pooled sensitivity was 0.44 (95% CI: 0.38–0.49), and pooled specificity was 0.96 (95% CI: 0.95–0.98). The SROC curve shown in Fig 4 yielded an AUC of approximately 0.90, indicating high accuracy in identifying individuals without kidney disease. However, the low sensitivity suggests that SUN may miss up to 56% of true cases when sCr is used as the reference.

The DOR was 18.86 (95% CI: 15.19–23.57), indicating moderate overall diagnostic performance. Additional diagnostic metrics, including likelihood ratios, are detailed in S5 Table (Section A). Due to the small number of studies (n = 2), heterogeneity statistics could not be estimated via the bivariate model and should be interpreted cautiously [20].

### Saliva urea nitrogen performance using blood urea nitrogen as the reference test

Five studies (Silva 2014, Silva 2018, Evans 2017, Raimann 2011, Raimann 2016) used BUN as the reference. As shown in Fig 5, pooled sensitivity was 0.83 (95% CI: 0.69–0.91), and specificity was 0.88 (95% CI: 0.78–0.94). The AUC from the SROC curve shown in Fig 6 was 0.93 (95% CI: 0.90–0.95), indicating high diagnostic accuracy. Compared to the sCr subgroup, the BUN subgroup showed higher sensitivity, suggesting SUN is more effective in detecting kidney dysfunction using BUN as a reference. However, a sensitivity of 83%, implies up to 17% of cases may still be missed.

The DOR for this subgroup was 37 (95% CI: 15–91), indicating strong discriminatory power. Detailed estimates are provided in S5 Table (Section B). Significant heterogeneity was observed with univariate I² values of 76.8% for sensitivity and 79.2% for specificity. However, when assessed using the bivariate model, the generalized inconsistency statistic

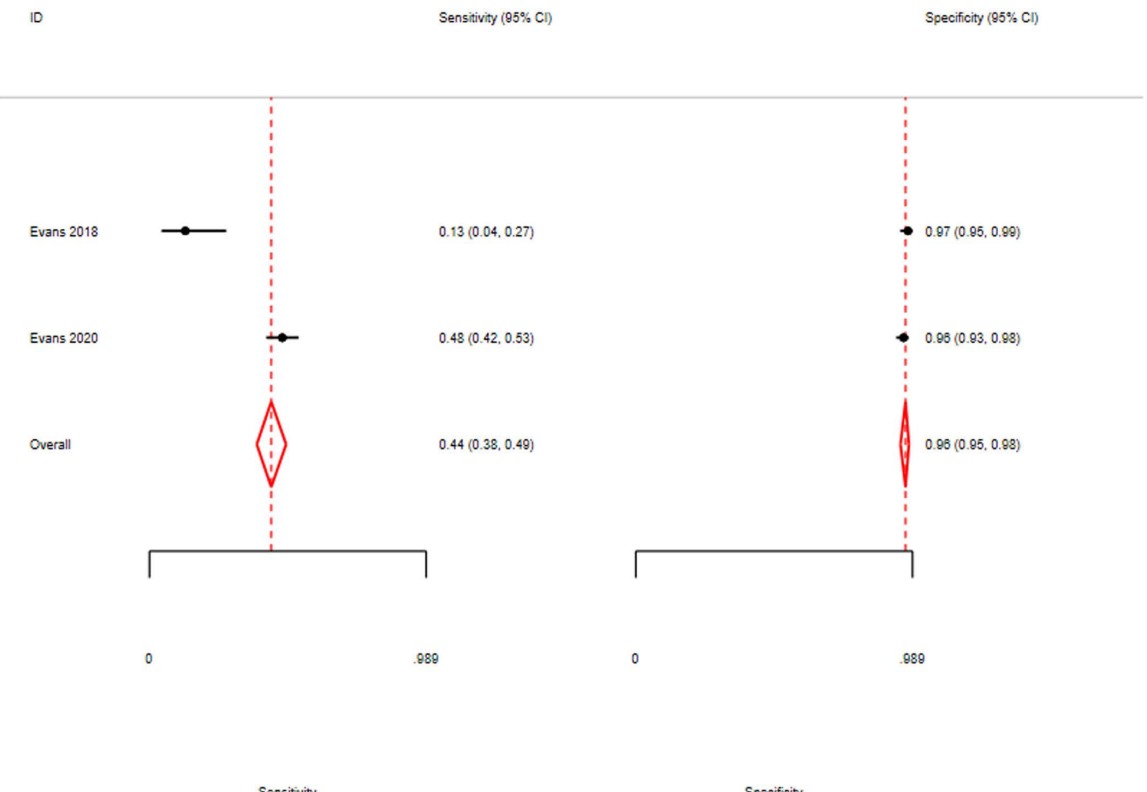

**Fig 3. Forest plot of pooled sensitivity and specificity for SUN using sCr as reference.**

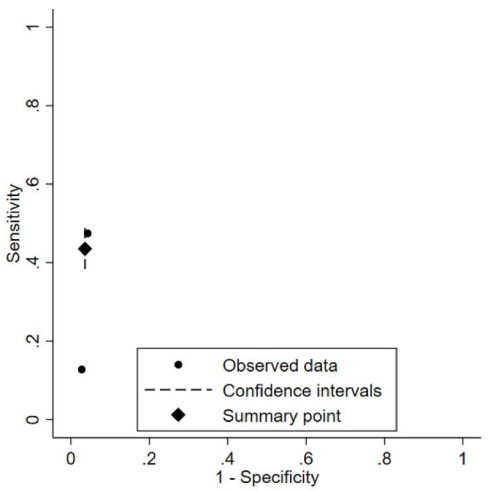

Note: Each circle represents one study.

**Fig 4. SROC curve for SUN using sCr as reference.**

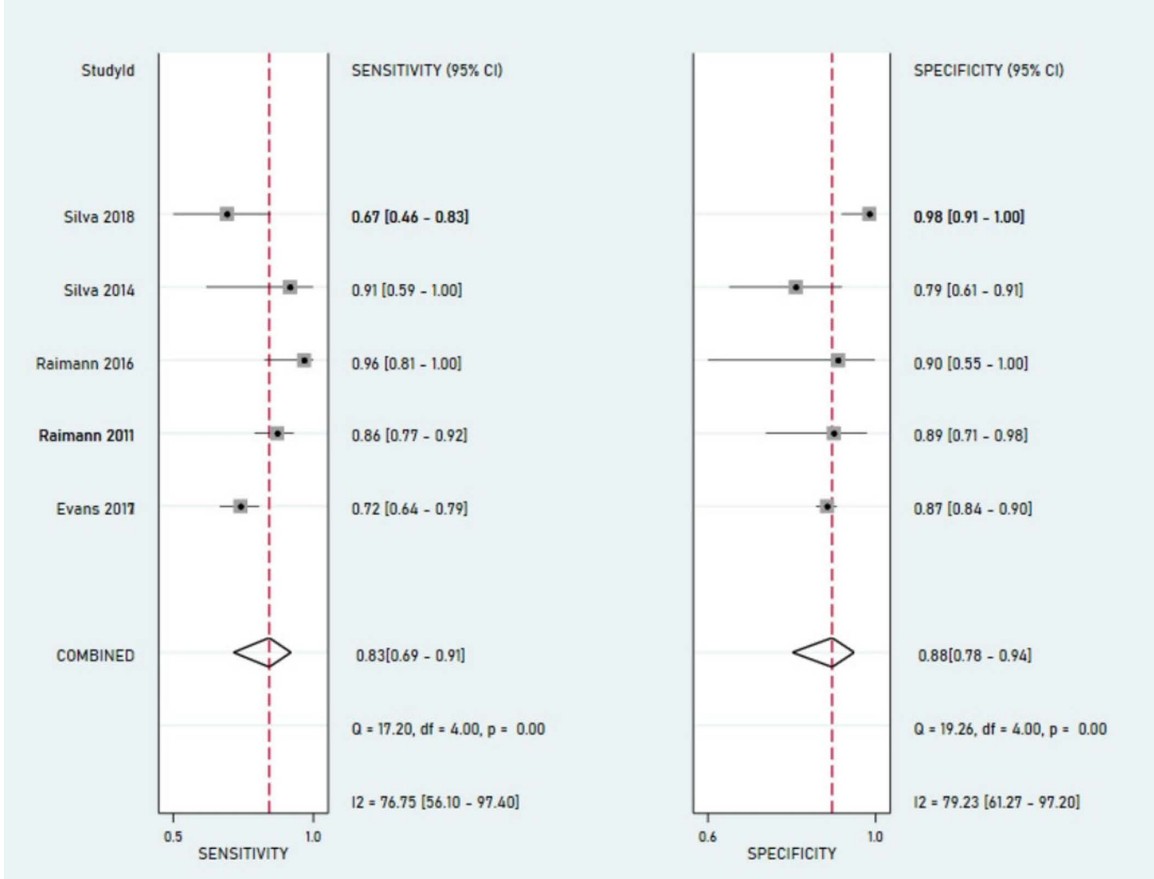

**Fig 5. Forest plot of pooled sensitivity and specificity for SUN using BUN as reference.**

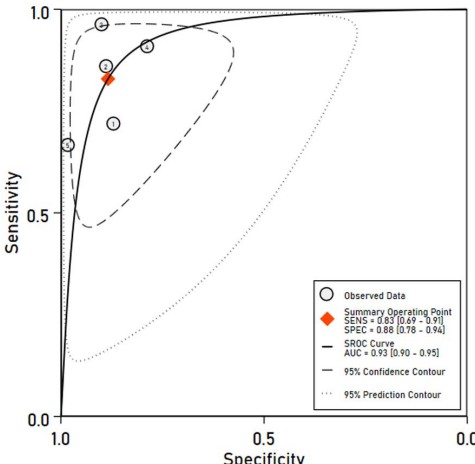

**Fig 6. SROC curve for SUN using BUN as reference.**

(LRT I²) was 51% with a non-significant chi-square test for heterogeneity ($\chi^2 = 4.08$, df = 2, p = 0.065), indicating moderate between-study variability. Threshold effects accounted for 42% of this heterogeneity. Additionally, median heterogeneity values were 0.66 for sensitivity and 0.63 for specificity. Due to the small number sCr studies (n = 2), advanced modelling and SROC contouring were limited. The BUN subgroup allowed more robust synthesis and visual analysis.

## Meta-regression and sources of heterogeneity

Although diagnostic performance was stratified by reference test, meta-regression was conducted across all seven studies to explore broader sources of heterogeneity. Due to the small number of studies in each subgroup (n = 2 for sCr, n = 5 for BUN), combined meta-regression was performed with "reference test used" included as a covariate-an approach consistent with Cochrane guidelines for DTA reviews [20]. As shown in Fig 7 and Table 3, meta-regression identified that study country (p = 0.01) and reference test used (p = 0.02) significantly contributed to heterogeneity in sensitivity, while comorbidity significantly (p = 0.001) influenced specificity.

## Discussion

This meta-analysis evaluated the diagnostic performance of SUN for detecting kidney disease, with analyses stratified by the reference test used (sCr or BUN). Findings suggest that SUN may serve as a promising non-invasive biomarker, particularly when benchmarked against BUN. In the BUN subgroup, SUN demonstrated a pooled sensitivity of 83% and specificity of 88%, with a DOR of 37, reflecting strong overall discriminatory power. In contrast, the sCr subgroup showed lower sensitivity (44%) but higher specificity (96%), with a DOR of 18.86. These findings indicate that while SUN is more

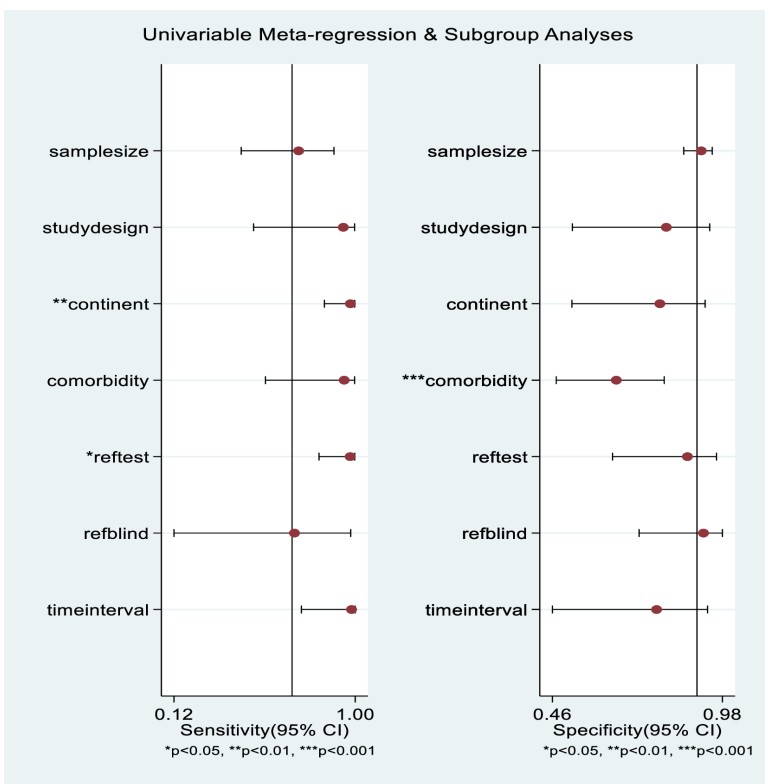

**Fig 7. Meta-regression analysis for sensitivity and specificity.**

**Table 3. Meta-regression results for SUN's diagnostic performance.**

| Parameter | Number of studies | Sensitivity (95% CI) | P1 | Specificity (95% CI) | P2 |
|---|---|---|---|---|---|
| Sample size | 7 | 0.72 (0.44–0.90) | 1.00 | 0.92 (0.86–0.95) | 1.00 |
| Study design | 7 | 0.94 (0.50 - 1.00) | 0.19 | 0.81 (0.52–0.94) | 0.29 |
| Study country | 7 | 0.98 (0.85 - 1.00) | **0.01** | 0.79 (0.52–0.93) | 0.19 |
| Comorbidity | 7 | 0.95 (0.56 - 1.00) | 0.16 | 0.65 (0.47–0.80) | 0.00 |
| Reference test | 7 | 0.97 (0.82 -1.00) | **0.02** | 0.87 (0.64–0.96) | 0.55 |
| Reference test blinded | 7 | 0.70 (0.12–0.98) | 0.97 | 0.92 (0.72–0.98) | 0.80 |
| Time interval | 7 | 0.98 (0.74. - 1.00) | 0.06 | 0.78 (0.46–0.94) | 0.21 |

effective in detecting true positive cases when compared to BUN, it performs better in ruling out disease when compared to sCr.

Compared to established biomarkers for kidney disease such as sCr and BUN, SUN presents several practical and clinical advantages. Although sCr is widely used, its sensitivity varies depending on disease stage and may fail to detect early dysfunction. While SUN's sensitivity is moderate, it aligns with sCr in early-stage detection, although, sCr tends to be more reliable in advanced stages [30]. BUN, however, is influenced by non-renal factors like hydration and dietary protein intake, which may reduce its diagnostic accuracy [31].

Cystatin C, another emerging biomarker, demonstrates higher sensitivity and specificity across all disease stages [32]. A previous meta-analysis of 24 studies (n= with 2, 007) reported DORs of 3.99 for Cystatin C and 2.79 for sCr [30]. In contrast, the DOR of of 37 observed for SUN in the BUN subgroup suggests potentially greater discriminatory power. However, comparisons across different meta-analytic models should be interpreted with caution. These results indicate that SUN, while not a replacement for existing biomarkers, may serve as a valuable adjunct, especially in low-resource settings where non-invasive alternatives are critical.

Beyond diagnostic performance, SUN offers notable practical advantages. The test is non-invasive, easy to administer, requires minimal training, and provides rapid results without the need for complex laboratory infrastructure. These features support its potential as a scalable, point-of-care tool in resource- limited settings [33].

In a separate meta-analysis of salivary biomarkers, salivary creatinine and salivary urea demonstrated higher sensitivity and overall accuracy than SUN. While SUN had lower sensitivity, it showed higher specificity, suggesting greater value in ruling out the disease [23]. While SUN had lower sensitivity, it showed higher specificity, suggesting greater value in ruling out disease. This trade-off may be clinically advantageous in triaging or surveillance contexts where reducing false positives is a priority.

Several confounding factors may influence SUN' diagnostic accuracy. Comorbid conditions such as malaria and obstetric-related kidney disease can affect kidney function and the SUN levels, complicating interpretation [16,29]. The choice of reference test (BUN vs. sCr), variation in diagnostic thresholds, inconsistent cut-off values likely contribute to observed heterogeneity and limit cross-study comparability [34].

Geographic variation in kidney disease prevalence, healthcare infrastructure, and population characteristics also affect SUN's diagnostic performance across studies [26,35]. Limited representation of diverse ethnic groups and common comorbidities such as diabetes and hypertension, further restricts generalizability. Additionally, inaccurate urine output measurements in some settings may have affected reference classifications. Smal study sizes in studies also introduce bias and impact pooled estimates [36–37].

Although this meta-analysis focused on SUN's role in diagnosing kidney disease, its utility may extend to other clinical contexts involving altered renal perfusion or urea metabolism, such as sepsis, dehydration, or heart failure. Future studies should explore SUN's potential as a general marker of renal or systemic stress in such settings. Standardization of SUN

measurement protocols, definition diagnostic thresholds, and large-scale studies in diverse populations are necessary to validate SUN's broader applicability and support its integration as a point-of-care diagnostic tool.

## Study limitations

While this meta-analysis provides preliminary insights into the diagnostic utility of SUN, several limitations must be acknowledged. First, only seven studies met the inclusion criteria, limiting the precision and statistical power of pooled estimates [38]. Three studies were conducted in the same country (Malawi), and five included at least one common author, potentially introducing author or site-related bias. However, this consistency may provide focused insight into SUN's performance in low-resource settings where it may be most impactful.

Study populations were relatively homogenous, with limited representation of different ethnicities and comorbidities such as diabetes or hypertension. These variables may influence SUN levels and diagnostic thresholds, underscoring the need for broader studies. Additionally, challenges in accurately measuring urine output in some settings may have affected the diagnostic classifications in the original studies.

The generalizability of findings to other regions, including broader sub-Saharan Africa and other low- and middle-income countries, must be interpreted cautiously given variations in healthcare access and clinical practices. Despite these limitations, this meta-analysis addresses a significant gap by quantitatively synthesizing existing evidence on a low-cost, non-invasive diagnostic tool with global health relevance. Further research should prioritize broader geographic representation, diverse populations, and standardized protocols.

## Conclusion

This meta-analysis provides preliminary evidence that SUN may serve as a practical, non-invasive biomarker for kidney disease, particularly due to its high specificity and strong performance in BUN-based comparisons. While heterogeneity and methodological limitations exist, the findings support SUN'S potential as an accessible tool in resource-limited settings. The results form a foundation for future validation studies, including threshold optimization and integration with other point-of-care diagnostics. Further research in diverse populations and clinical settings is essential to confirm SUN's utility and promote its broader implementation.

## Supporting information

**S1 Table. PRISMA-DTA checklist.**
(DOCX)

**S2 Table. Detailed search strategy for each database.**
(DOCX)

**S1 File. List of screened studies with inclusion status and reasons for exclusion.**
(XLSX)

**S2 File. Extracted data from included primary studies.**
(XLSX)

**S3 Table. Risk of bias and quality/certainly assessments for included studies.**
(DOCX)

**S4 Table. Diagnostic test performance of individual studies that used saliva urea nitrogen.** PPV: positive predictive value; NPV: negative predictive value; LR + : positive likelihood ratio; LR-: negative likelihood ratio; DOR: diagnostic odds ratio; CI: Confidence Interval.
(DOCX)

**S5 Table. Diagnostic accuracy measures for sCr and BUN subgroup analyses (with 95% confidence intervals).** (DOCX)

## Acknowledgments

The authors express their appreciation to the participants and researchers involved in the original studies included in this review. We also extend our gratitude to IMU for providing the support necessary to conduct this study.

## Author contributions

**Conceptualization:** Mona Mohamed Ibrahim Abdalla, Reyna Kumaran.

**Data curation:** Mona Mohamed Ibrahim Abdalla, Reyna Kumaran, Brinnell Annette Caszo, Sushela Devi Somanath.

**Formal analysis:** Mona Mohamed Ibrahim Abdalla.

**Funding acquisition:** Mona Mohamed Ibrahim Abdalla, Reyna Kumaran, Brinnell Annette Caszo, Sushela Devi Somanath.

**Supervision:** Mona Mohamed Ibrahim Abdalla, Brinnell Annette Caszo, Sushela Devi Somanath.

**Visualization:** Mona Mohamed Ibrahim Abdalla.

**Writing – original draft:** Mona Mohamed Ibrahim Abdalla, Reyna Kumaran.

**Writing – review & editing:** Mona Mohamed Ibrahim Abdalla, Reyna Kumaran, Brinnell Annette Caszo, Sushela Devi Somanath.

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
