## [Decision Letter · Decision Letter 0]

13 Feb 2025

PONE-D-25-02263Saliva urea nitrogen for detection of kidney disease in adults: A meta-analysis of diagnostic test accuracyPLOS ONE

Dear Dr. Abdalla,

Thank you for submitting your manuscript to PLOS ONE. After careful consideration, we feel that it has merit but does not fully meet PLOS ONE’s publication criteria as it currently stands. Therefore, we invite you to submit a revised version of the manuscript that addresses the points raised during the review process. Please submit your revised manuscript by Mar 30 2025 11:59PM. If you will need more time than this to complete your revisions, please reply to this message or contact the journal office at plosone@plos.org . Please include the following items when submitting your revised manuscript:

We look forward to receiving your revised manuscript.

Kind regards,

Phakkharawat Sittiprapaporn, Ph.D.

Academic Editor

PLOS ONE

“International Medical University (IMU) in Malaysia [ID: CSc-Sem6(08)2022]”

Additional Editor Comments:

Some suggestions are made as follows to improve the quality of the manuscript.

(a) Recent studies have identified several novel biomarkers associated with kidney illnesses. To make the manuscript better, the introduction should include summaries of more earlier studies.

(b) To do the meta-regression, the author(s) you need to figure out the sensitivity and specificity of SUN for each of the two reference tests individually. The analyses should come up with pooled figures for each test, not for the whole thing.

(c) The use of only a few literature studies raises major questions about the usefulness of the given meta-analysis. Furthermore, it's important to outline the limitations.

(d) The language editing needs improvement by a native speaker.

Reviewers' comments:

Reviewer's Responses to Questions

**Comments to the Author**

1. Is the manuscript technically sound, and do the data support the conclusions?

Reviewer #1: Yes

Reviewer #2: Partly

Reviewer #3: No

Reviewer #4: Partly

2. Has the statistical analysis been performed appropriately and rigorously? 

Reviewer #1: Yes

Reviewer #2: No

Reviewer #3: Yes

Reviewer #4: No

3. Have the authors made all data underlying the findings in their manuscript fully available?

Reviewer #1: Yes

Reviewer #2: Yes

Reviewer #3: Yes

Reviewer #4: Yes

4. Is the manuscript presented in an intelligible fashion and written in standard English?

Reviewer #1: No

Reviewer #2: Yes

Reviewer #3: Yes

Reviewer #4: Yes

5. Review Comments to the Author

Reviewer #1: In this work, the authors reveal that saliva urea nitrogen for detection of kidney disease in adults: A meta-analysis of diagnostic test accuracy. Several suggestions are made as follows to improve the quality of the manuscript.

1. All abbreviations should be substantiated for the first time.

2. Several novel biomarkers were identified in renal diseases by the latest studies such as PMID: 38538857; PMID: 39098923; PMID: 38053242; PMID: 35577910; PMID: 26376947. Please summarize these previous studies in the introduction section to improve manuscript.

3. Publishing clinical studies should be further discussed.

4. Limitations should be described.

5. Please change the references based on the guide for authors.

6. The language editing should be improved by a native speaker.

Reviewer #2: The author has reported a review paper regarding the current status of Salivary Urea nitrogen tests for screening of chronic kidney disease. The manuscript is well written and consists of important information. Although there are few points which needs to be addressed for better readability of this work, mentioned pointwise:

1. Line no. 71: SUN is the abbreviation of "Salivary urea nitrogen", use this phrase instead.

2. The introduction does not depict the level of SUN for healthy and sick, the information should be there.

3. Comparative analysis of alternative biomarkers such as Scr and BUN is missing. Author's claim on SUNs performance is vague until it is benchmarked with Scr and BUN

4. What is the key takeaways of this study? How experimental research will get benefit from this study?

5. Author should consider experimental factors of SUN as success criterion too.

6. How SUN model will be effective with other non specific disease state along with CKD?

7. Conclusion is not technical, need to be revised.

Reviewer #3: The manuscript provides a meta-analysis of a diagnostic test accuracy: the use of measurements of saliva urea nitrogen (SUN) as a method for detection of kidney disease in undeveloped countries. The manuscript is well written – no particular issues with language detected; each section of the manuscript is also comprehensively presented.

Major concern/comment:

I have a major concern in relation to the merit of the presented meta-analysis on the exclusive basis of the low number of literature studies used. After all the search made and through the application of the eligibility criteria only seven studies remained, references 18-24. From these studies three are made in the same country (Malawi) and all seven share at least one author in the list of authors of the publication. One of the authors (Rhys Evans) is present in five of the seven publications. Thus, the question to be raised is why did the authors decided to perform such meta-analysis considering these facts?

Reviewer #4: I read the meta-analysis on SUN with great interest. The paper is well written, and the importance of the work is clearly and compellingly communicated. The authors address an important gap in the literature and there is value in conducting a meta-analysis to aggregate published studies. My main concerns when reading the paper was switching back and forth between reference tests presenting sensitivity or specificity for certain tests without presenting both sets of data for both reference tests. Further, the thresholds for the reference tests are not provided. The authors should clearly indicate what the reference test is, and if they are going to include two, then specify which is primary and secondary, and present all statistics for each test. Further, the patient population should be better defined. It is difficult to conceptualize using the test for both CKD, AKD, and AKI and using BUN and creatinine as reference tests.

Specific comments

For the meta-regression, the sensitivity and specificity of SUN need to be calculated separately for the two reference tests. It’s also hard to envision how estimates can be pooled when thresholds of what constitutes a positive test differs across studies. The analyses should generate pooled estimates within each test, not overall.

The reference test varies between SCr and BUN but in both cases, the thresholds of what constitutes a positive reference test is not provided and is needed.

For the sources of heterogeneity, consider the threshold of the tests used both for the index and the reference test.

The paragraph on a previous meta-analysis on salivary creatinine and urea is interesting and the data appropriately contextualized given the results presented. This needs to be reconsidered when the estimates are provided for each reference test separately, as the interpretation is likely to change.

6. PLOS authors have the option to publish the peer review history of their article (what does this mean? ). If published, this will include your full peer review and any attached files.

**Do you want your identity to be public for this peer review?** For information about this choice, including consent withdrawal, please see our Privacy Policy .

Reviewer #1: No

Reviewer #2: No

Reviewer #3: No

Reviewer #4: No

---

## [Author Response · Author response to Decision Letter 1]

15 Apr 2025

Answers point by point is included in the file "Response to Reviewers' Comments". Response to Reviewers

Dear editor and reviewers,

Thank you for your valuable feedback on our manuscript. We really appreciate your input, and the time spent in review. We have carefully addressed all the points raised in the revised manuscript. The table below shows the response to each point.

Comments- Editor

https://journals.plos.org/plosone/s/file?id=wjVg/PLOSOne_formatting_sample_main_body.pdf andhttps://journals.plos.org/plosone/s/file?id=ba62/PLOSOne_formatting_sample_title_authors_affiliations.pdf

Response: The manuscript has been formatted according to PLOS ONE style

“International Medical University (IMU) in Malaysia [ID: CSc-Sem6(08)2022]”

Response: The role of funder statement has been included in the cover letter "The funders had no role in study design, data collection and analysis, decision to publish, or preparation of the manuscript."

Additional Editor Comments:

Some suggestions are made as follows to improve the quality of the manuscript.

(a) Recent studies have identified several novel biomarkers associated with kidney illnesses. To make the manuscript better, the introduction should include summaries of more earlier studies.

Response: We have reviewed the studies corresponding to the provided PMIDs and integrated a concise summary of these novel biomarkers into the Introduction section to enhance the context and relevance of our study.

(b) To do the meta-regression, the author(s) you need to figure out the sensitivity and specificity of SUN for each of the two reference tests individually. The analyses should come up with pooled figures for each test, not for the whole thing.

Response: In line with the editor’s recommendation, we stratified our meta-analysis based on the reference standard used (BUN or sCr), and separately calculated pooled sensitivity, specificity, and AUC for each subgroup. This stratified approach enables a clearer interpretation of SUN’s performance in different diagnostic contexts.

(c) The use of only a few literature studies raises major questions about the usefulness of the given meta-analysis. Furthermore, it's important to outline the limitations.

Response: Regarding the limited number of studies, we acknowledge this as a key limitation and have expanded the Limitations section accordingly. Despite the small sample, we conducted this meta-analysis to offer the first quantitative synthesis of SUN's diagnostic accuracy, an emerging, low-cost, and non-invasive biomarker with growing interest, particularly in low-resource settings. Our findings serve as a foundation for future studies to build upon and validate.

d) The language editing needs improvement by a native speaker.

Response: The revised manuscript has been thoroughly revised by a native speaker.

Response: We have completed the requested step. The figure files have been processed through the PACE tool as instructed, and the PACE-validated versions have been uploaded along with the revised manuscript.

Reviewer 1 Comments

1. All abbreviations should be substantiated for the first time.

Response: We have carefully reviewed the manuscript and ensured that all abbreviations are fully defined upon first mention.

2. Several novel biomarkers were identified in renal diseases by the latest studies such as PMID: 38538857; PMID: 39098923; PMID: 38053242; PMID: 35577910; PMID: 26376947. Please summarize these previous studies in the introduction section to improve manuscript.

3. Publishing clinical studies should be further discussed.

Response: In response, we reviewed the studies corresponding to the provided PMIDs and have incorporated a summary of the key findings into the Introduction section to enhance the background and relevance of our work. These have been appropriately cited (references 7–10). Additionally, we have expanded the discussion around clinical studies to better contextualize the role of salivary biomarkers in diagnostic research.

4. Limitations should be described.

Response: The Limitations section has been revised to clearly acknowledge the study’s constraints, including the limited number of included studies and heterogeneity among reference standards and thresholds.

5. Please change the references based on the guide for authors.

Response: All references have been reformatted in accordance with the PLOS ONE author guidelines.

6. The language editing should be improved by a native speaker.

Response: The manuscript has undergone comprehensive language editing by a native English speaker to ensure clarity, fluency, and academic tone.

Reviewer 2- Comments

1. Line no. 71: SUN is the abbreviation of "Salivary urea nitrogen", use this phrase instead.

Response: Revised accordingly.

2. The introduction does not depict the level of SUN for healthy and sick, the information should be there.

Response: The Introduction section has been updated to include typical SUN levels in both healthy individuals and those with kidney disease. Specifically: “SUN levels in healthy individuals usually range from 5 to 14 mg/dL, while levels above 35–54 mg/dL are commonly observed in kidney dysfunction.”

3. Comparative analysis of alternative biomarkers such as Scr and BUN is missing. Author's claim on SUNs performance is vague until it is benchmarked with Scr and BUN

Response: We appreciate this observation. The Discussion has been expanded to compare SUN with established biomarkers like serum creatinine blood urea nitrogen, and cystatin C. We included comparative metrics such as diagnostic odds ratios , and highlighted SUN’s strengths as a non-invasive, saliva-based option, especially useful in low-resource settings.

4. What is the key takeaways of this study? How experimental research will get benefit from this study?

Response: We have clarified the study's key takeaways in the Conclusion. This meta-analysis confirms that SUN is a reliable, non-invasive tool for detecting kidney disease. The summarized diagnostic metrics (sensitivity, specificity, AUC) offer a benchmark for future experimental research to design validation studies, optimize thresholds, and explore SUN in combination with other point-of-care tools, particularly in underserved settings.

5. Author should consider experimental factors of SUN as success criterion too.???

Response: Agreed. We have added a paragraph in the Discussion emphasizing SUN’s practical strengths, including its simplicity, low cost, non-invasiveness, and feasibility in remote or resource-limited environments. These factors support its potential for broader clinical and public health application.

6. How SUN model will be effective with other non specific disease state along with CKD?

Response: Thank you for this valuable point. We expanded the Discussion to mention that SUN levels may also change in conditions like dehydration, sepsis, liver dysfunction, and heart failure, due to shifts in protein metabolism or renal perfusion. While further research is needed, SUN may hold potential as a general indicator of renal and systemic stress, especially when used alongside other non-invasive markers.

7. Conclusion is not technical, need to be revised.

Response: Thank you for the helpful feedback. In response, we have revised the Conclusion to adopt a more technical tone. It now emphasizes key diagnostic metrics, highlights SUN’s clinical potential as a non-invasive biomarker, and outlines the implications for future validation studies and broader application in diverse and resource-limited settings.

Reviewer 3- Comments

Major concern/comment:

I have a major concern in relation to the merit of the presented meta-analysis on the exclusive basis of the low number of literature studies used. After all the search made and through the application of the eligibility criteria only seven studies remained, references 18-24. From these studies three are made in the same country (Malawi) and all seven share at least one author in the list of authors of the publication. One of the authors (Rhys Evans) is present in five of the seven publications. Thus, the question to be raised is why did the authors decided to perform such meta-analysis considering these facts?

Response: We acknowledge the reviewer’s concern regarding the small number of studies, geographic concentration, and overlapping authorship. These limitations are now clearly reflected in the revised manuscript. However, given the growing interest in SUN as a low-cost, non-invasive diagnostic tool, especially in low-resource settings, and the absence of any prior meta-analysis on this topic, we believe it is timely to synthesize the available evidence. Our study provides preliminary pooled estimates, identifies key gaps, and lays a foundation for future, more diverse validation efforts. We hope it serves as a starting point to stimulate further research in this area.

Reviewer 4- Comments

I read the meta-analysis on SUN with great interest. The paper is well written, and the importance of the work is clearly and compellingly communicated. The authors address an important gap in the literature and there is value in conducting a meta-analysis to aggregate published studies. My main concerns when reading the paper was switching back and forth between reference tests presenting sensitivity or specificity for certain tests without presenting both sets of data for both reference tests. Further, the thresholds for the reference tests are not provided. The authors should clearly indicate what the reference test is, and if they are going to include two, then specify which is primary and secondary, and present all statistics for each test. Further, the patient population should be better defined. It is difficult to conceptualize using the test for both CKD, AKD, and AKI and using BUN and creatinine as reference tests.

Response: We thank the reviewer for the thoughtful feedback. In response, we have clarified which reference test (BUN or sCr) is primary and now present sensitivity and specificity separately for each. Diagnostic thresholds are detailed in the revised Table 2. We also refined the description of patient populations (CKD, AKD, AKI) in Table 1 and added a note explaining how reference test selection aligns with clinical context. These updates are reflected in the stratified analysis and discussed as sources of heterogeneity, highlighting the need for standardization in future research.

1. For the meta-regression, the sensitivity and specificity of SUN need to be calculated separately for the two reference tests. It’s also hard to envision how estimates can be pooled when thresholds of what constitutes a positive test differs across studies. The analyses should generate pooled estimates within each test, not overall.

Response: In response, we have clarified which reference test (BUN or sCr) is primary and now present sensitivity and specificity separately for each. The analyses now generate pooled estimates within each test, not overall.

2. The reference test varies between SCr and BUN but in both cases, the thresholds of what constitutes a positive reference test is not provided and is needed.

Response: The thresholds used to define a positive result for each reference test are now detailed in Table 2.

3. For the sources of heterogeneity, consider the threshold of the tests used both for the index and the reference test.

Response: We have accounted for variations in diagnostic thresholds as a potential source of heterogeneity. This has been analyzed and discussed in the Results and Discussion sections.

4. The paragraph on a previous meta-analysis on salivary creatinine and urea is interesting and the data appropriately contextualized given the results presented. This needs to be reconsidered when the estimates are provided for each reference test separately, as the interpretation is likely to change.

Response: The paragraph has been revised to reflect the stratified analysis. Interpretations have been updated accordingly to align with the revised sensitivity and specificity results for each reference test.

---

## [Decision Letter · Decision Letter 1]

23 Apr 2025

Saliva urea nitrogen for detection of kidney disease in adults: A meta-analysis of diagnostic test accuracy

PONE-D-25-02263R1

Dear Dr. Abdalla,

We’re pleased to inform you that your manuscript has been judged scientifically suitable for publication and will be formally accepted for publication once it meets all outstanding technical requirements.

Kind regards,

Assoc. Prof. Phakkharawat Sittiprapaporn, Ph.D.

Academic Editor

PLOS ONE

Additional Editor Comments (optional):

Reviewers' comments:

Reviewer's Responses to Questions

**Comments to the Author**

1. If the authors have adequately addressed your comments raised in a previous round of review and you feel that this manuscript is now acceptable for publication, you may indicate that here to bypass the “Comments to the Author” section, enter your conflict of interest statement in the “Confidential to Editor” section, and submit your "Accept" recommendation.

Reviewer #1: All comments have been addressed

Reviewer #3: All comments have been addressed

2. Is the manuscript technically sound, and do the data support the conclusions?

Reviewer #1: Yes

Reviewer #3: Yes

3. Has the statistical analysis been performed appropriately and rigorously? 

Reviewer #1: Yes

Reviewer #3: Yes

4. Have the authors made all data underlying the findings in their manuscript fully available?

Reviewer #1: Yes

Reviewer #3: Yes

5. Is the manuscript presented in an intelligible fashion and written in standard English?

Reviewer #1: Yes

Reviewer #3: Yes

6. Review Comments to the Author

Reviewer #1: After careful revision according to the comments of reviewer, the manuscript has been improved as compared to previous version, it is recommended for publication.

Reviewer #3: I am fine with the author's reply to my comment and all other reviewer comments. The new version of the manuscript has been considerably improved in overall quality.

7. PLOS authors have the option to publish the peer review history of their article (what does this mean? ). If published, this will include your full peer review and any attached files.

**Do you want your identity to be public for this peer review?** For information about this choice, including consent withdrawal, please see our Privacy Policy .

Reviewer #1: No

Reviewer #3: **Yes: ** Rui de Albuqerque Carvalho

---

## [Editor Report · Acceptance letter]

PONE-D-25-02263R1

PLOS ONE

Dear Dr. Abdalla,

I'm pleased to inform you that your manuscript has been deemed suitable for publication in PLOS ONE. Congratulations! Your manuscript is now being handed over to our production team.

Kind regards,

on behalf of

Assoc. Prof. Dr. Phakkharawat Sittiprapaporn

Academic Editor

PLOS ONE